# Building on Foundations: Venetoclax-Based Combinations in the Treatment of Acute Myeloid Leukemia

**DOI:** 10.3390/cancers15143589

**Published:** 2023-07-12

**Authors:** Emmanuella Oyogoa, Elie Traer, Jeffrey Tyner, Curtis Lachowiez

**Affiliations:** 1Department of Internal Medicine, Oregon Health & Science University, Portland, OR 97239, USA; oyogoa@ohsu.edu; 2Knight Cancer Institute, Division of Hematology/Medical Oncology, Oregon Health & Science University, Portland, OR 97239, USA; 3Department of Cell, Developmental & Cancer Biology, Oregon Health & Science University, Portland, OR 97239, USA

**Keywords:** acute myeloid leukemia, venetoclax, combined targeted therapy, hypomethylating agent

## Abstract

**Simple Summary:**

Treatment of acute myeloid leukemia continues to progress. Although a number of therapies are efficacious in treating acute myeloid leukemia, some are intolerable to older patients or patients who have more advanced disease. In light of this, newer therapies that are both efficacious and tolerable are being studied as a monotherapy and in combination with other therapies. In this review article, we aim to summarize on going clinical trials of venetoclax in combination with other targeted therapies. When combined with other therapies, venetoclax demonstrates synergy in comparison to venetoclax monotherapy or alternate therapy alone. In addition, venetoclax combinations appear to overcome a number of known resistance mechanisms to venetoclax.

**Abstract:**

Frontline acute myeloid leukemia (AML) treatment is determined by a combination of patient and genetic factors. This includes patient fitness (i.e., comorbidities that increase the risk of treatment-related mortality) and genetic characteristics, including cytogenetic events and gene mutations. In older unfit patients, the standard of care treatment is typically venetoclax (VEN) combined with hypomethylating agents (HMA). Recently, several drugs have been developed targeting specific genomic subgroups of AML patients, enabling individualized therapy. This has resulted in investigations of doublet and triplet combinations incorporating VEN aimed at overcoming known resistance mechanisms and improving outcomes in older patients with AML. These combinations include isocitrate dehydrogenase-1/2 (IDH1/2) inhibitors (i.e., ivosidenib and enasidenib), fms-like tyrosine kinase 3 (FLT3) inhibitors (i.e., gilteritinib), anti-CD47 antibodies (i.e., magrolimab), mouse double minute-2 (MDM2) inhibitors, and p53 reactivators (i.e., eprenetapopt). This review summarizes ongoing trials aimed at overcoming known VEN resistance mechanisms and improving outcomes beyond that observed with HMA + VEN combinations in the treatment of AML.

## 1. Introduction

Treatment for fit patients with AML generally utilizes intensive chemotherapy (IC) which incorporates a continuous infusion of cytarabine (100–200 mg/m^2^) for 7 days combined with 3 days of daunorubicin (60–90 mg/m^2^; the so-called “7 + 3” regimen) [1]. In younger patients (<age 60) without significant comorbidities, response rates range from 60 to 80% [2]. Patients aged ≥60 years and/or those with comorbidities treated with 7 + 3 experience increased rates of treatment-related mortality compared to their younger counterparts [3] and were previously treated with less intensive therapy, including azacitidine (AZA) or decitabine (DAC)AZA monotherapy [4]. 

A study by Kantarjian et al. evaluated the use of IC in older patients [3]. In patients aged 70 or older receiving IC, the 4-week and 8-week mortality was 26% and 36%, respectively. The overall complete response rate was 45%. One-year overall survival (OS) was 28%, and the median OS was 4.6 months. Thus, IC in older unfit patients with AML is associated with a poor prognosis, in part due to an increased incidence of underlying adverse disease biology, including unfavorable karyotype, secondary AML, and therapy-related AML. Similarly, in patients aged 60 and over with AML receiving IC, outcomes were poor, with low survival rates and decreased CR rates [5]. In patients aged 75 or older (or those considered unfit for conventional IC), the hypomethylating agents (AZA or DAC) combined with the BCL2 inhibitor VEN are now standard of care [4,6], significantly improving OS compared to AZA monotherapy (median 14.7 vs. 9.6 months) [4]. In addition, the combination of low-dose cytarabine (LDAC) and VEN is an alternative therapy that is both efficacious and safe in ND-AML patients ineligible for IC [7].

Genetic profiling is integral for AML prognostication, as certain cytogenetic events or molecular mutations influence prognosis and response to treatment. This profiling has simultaneously enabled the identification of additional drug targets [8,9]. For example, internal tandem duplications within the fms, like the tyrosine kinase 3 gene (*FLT3*-ITD), are associated with inferior survival [10]. Similarly, the recognition of leukemogenic mutations in *IDH1* and *IDH2* led to the development of IDH1 (ivosidenib (IVO), olutasidenib (OLUTA)) and IDH2 (enasidenib (ENA)) inhibitors to treat patients with AML harboring the respectively mutated IDH1 or IDH2 gene [11,12,13]. 

Given the potential for synergistic efficacy when combining targeted therapies, numerous early-phase clinical trials combining VEN with targeted agents for the treatment of AML are ongoing. Herein, we review the clinical development and reported outcomes of VEN-based targeted combination therapies for the treatment of AML.

## 2. Venetoclax Mechanism of Action 

VEN, a BH3 mimetic, is a competitive antagonist of the anti-apoptotic B cell lymphoma 2 (BCL2) protein, a key survival protein in the intrinsic apoptotic pathway [14,15,16,17,18]. Given the pro-survival influence BCL2 imparts in AML, BCL2 emerged as an attractive therapeutic target. In a phase 1 study, the dual BCL2 and BCL2L1 (BCL-xL) inhibitor navitoclax demonstrated on-target activity, but with the off-tumor effect of thrombocytopenia [19]. Subsequently, VEN, a second-generation BCL2 inhibitor with greater affinity for BCL2 rather than BCL2L1, was developed. Accordingly, the degree of thrombocytopenia observed with VEN is less prominent than that observed with navitoclax [20]. 

## 3. Venetoclax as Monotherapy in AML

VEN was initially assessed in a single-arm phase 2 study, evaluating the activity of VEN monotherapy in relapsed refractory (R/R) AML. The study population included older adults (median age 74 years), of whom 94% had received at least one prior line of therapy. VEN resulted in an overall CR/CRi rate of 19% (CR: 6%, CRi: 13%). In patients with IDH1/2 mutations, 33% achieved CR/CRi, including one patient achieving CRi at week 24 after a 20-day dose interruption. The median duration of CR was 48 days [21]. VEN monotherapy has not been further tested in newly diagnosed AML (ND-AML). The phase 1b dose-escalation CAVEAT trial employed a 7-day VEN pre-phase prior to combining VEN with an attenuated “5 + 2” regimen of anthracycline and cytarabine-based chemotherapy. In this older patient population (median age 72 years), 72% attained CR/CRi (CR: 41%; CRi: 31%). Notable blast reductions occurred in several genomic subgroups following the 7-day VEN pre-phase, including patients with mutations in *NPM1* (median blast reduction 56%), *IDH2* (median blast reduction 55%), and *SRSF2* (median blast reduction 47%). After a median follow up of 22.9 months, the median OS was 11.2 months [22].

## 4. Venetoclax Resistance

Primary or acquired resistance to VEN or HMA + VEN combination therapy can occur via multiple mechanisms, many of which (including both cell maturation state and molecular aberrations) converge on increased alternative anti-apoptotic proteins, in particular MCL-1 and BCL2L1 [21,23,24]. Indeed, many VEN-resistant AML cell lines can be re-sensitized to VEN by targeting MCL-1 or BCL2L1 [25]. 

Molecular mechanisms also impart sensitivity or resistance to HMA + VEN combinations [26]. Acquisition or expansion of mutations in active signaling (i.e., *FLT3*-ITD, *N/KRAS*, and *PTPN11*) or tumor suppressor (i.e., *TP53*) genes correlate with acquired VEN resistance, while mutations in genes including *NPM1* or *IDH1/2* correlate with VEN sensitivity. Expanding *FLT3*-ITD mutated clones were noted at relapse following AZA + VEN, suggesting this molecular subgroup of patients may benefit from combining FLT3i with VEN to prevent relapse via FLT3-mediated mechanisms [27]. Similarly, targeting BCL2 and MAPK signaling using VEN combined with the MEK1 inhibitor cobimetinib resulted in synergistic growth inhibition in both VEN and cobimetinib-resistant cells [28]. 

Intact p53 protein function appears essential for sustained response to VEN. Expansion of *TP53*-mutated clones (often in conjunction with co-occurring signaling pathway mutations) is frequently observed in the context of *TP53*-mutated AML following VEN-based treatment. Notably, co-inhibition of BCL2 and MCL1 in vitro partially abrogated observed resistance in *TP53*-mutated AML, suggesting combined anti-apoptotic protein inhibition could be effective in this high-risk patient subgroup [29]. Recent work has demonstrated that AML differentiation state, in particular monocytic differentiation, can drive resistance to VEN [25]. Certain drugs appear more effective in monocytic AML when combined with VEN, demonstrating enhanced ex vivo activity dependent upon the maturation state [30]. The proposed mechanisms of both primary and acquired VEN resistance are depicted in Figure 1.

To complement pre-clinical models demonstrating combinatorial synergy of therapies targeting these aforementioned resistance pathways, multiple ongoing studies are investigating combining cellular and molecularly targeted therapies with VEN +/− AZA as doublet or triplet combinations as outlined below in Table 1, including several with published early outcomes data (Figure 2). 

## 5. Venetoclax Combined with Chemotherapy 

### 5.1. HMA + VEN

VEN combined with HMA (AZA or DAC) was investigated in a phase 1b (P1b) dose-escalation and expansion study of 145 untreated, elderly patients with AML exploring three different VEN dose levels (400 mg, 800 mg, and 1200 mg) [31]. The median age of the overall study population (dose escalation and dose expansion cohorts) was 74 (range 65–86); 62% of patients had an Eastern Cooperative Oncology Group Performance Status (ECOG PS) of 1. Forty-nine percent of patients had poor-risk cytogenetics. Common grade 3/4 adverse events (AE) encountered across the study population included febrile neutropenia (43%), leukopenia (31%), anemia (25%), thrombocytopenia (24%), neutropenia (17%), and pneumonia (13%). The overall response rate (ORR; CR + CRi + PR) was 68% in the intent-to-treat group, including a CR and CRi rate of 37% and 30%, respectively. Notably, CR/CRi rates were maintained across genomic subgroups (*NPM1*: 91%, *FLT3*-ITD/TKD: 72%, and *IDH1/2*: 71%). After a median follow up of 15.1 months, the median OS was 17.5 months.

In the confirmatory prospective, placebo-controlled, randomized phase 3 VIALE-A trial, AZA combined with VEN (AZA + VEN) was compared with AZA + placebo in the treatment of older (age ≥ 75 years) or unfit (i.e., unable to receive IC) adults with newly diagnosed (ND) AML. Akin to the findings in the P1b study, AZA + VEN resulted in a composite complete remission (CRc; CR + CRi) of 66.4% compared to 28.3% with AZA monotherapy. Notably, AZA + VEN resulted in improved rates of CRc across molecular subgroups, including 75.4% of patients with *IDH1/2* mutations, 66.7% of patients with *NPM1* mutations, 72.4% of patients with *FLT3*, and 55.3% of patients with *TP53* mutations [29]. After a median follow up of 20.5 months, AZA + VEN improved OS compared to AZA + placebo (median OS: 14.7 vs. 9.6 months, *p*-value: <0.001).

Most non-hematologic AEs were low grade, including nausea (AZA + VEN: 44% vs. AZA: 35%), constipation (43% vs. 39%), and diarrhea (41% vs. 33%). Common grade 3 or greater hematologic AEs included thrombocytopenia (AZA + VEN: 45% vs. AZA: 38%), neutropenia (42% and 28%), febrile neutropenia (42% and 19%; including 30% and 10% characterized as serious adverse events [SAEs], respectively). SAEs were observed in 83% of patients treated with AZA + VEN compared with 73% of patients treated with AZA.

Tumor lysis syndrome (TLS) occurring during days 1 through 3 during VEN dose-escalation was reported in 1% of patients. All patients were able to continue treatment, and TLS resolved with medical management and hydration, indicating that with judicious monitoring and fluid management, TLS risk can be mitigated with VEN in patients with AML. In light of these positive safety and efficacy results, AZA + VEN has become the standard of care for the treatment of older adults with AML [4].

### 5.2. IC + VEN 

VEN has also been evaluated in combination with IC, with promising early results. A phase Ib/II study evaluating the safety and efficacy of fludarabine, cytarabine, granulocyte colony-stimulating factor, and idarubicin in combination with VEN (FLAG-IDA + VEN) demonstrated high response rates and transition to successful transplant in patients with ND and R/R-AML [32]. A total of 68 patients were enrolled in the study. The median age of the study participants was 46 years (range 20–73). Across three study cohorts (PIb: R/R-AML (N = 16), PIIA: ND-AML (N = 29), PIIB: R/R-AML (N = 23)), the ORR was 75%, 97%, and 70%, respectively. Composite CR rates were 75% for P1b, 90% for PIIA, and 61% for PIIb participants. Among patients with ND-AML, 96% achieved MRD-negative remissions measured using multiparameter flow cytometry (MFC). At the recommended phase 2 dose, the combination of FLAG-IDA + VEN was safe with a manageable toxicity profile. Grade 3 or greater AEs recorded in >10% of patients included febrile neutropenia (50%), bacteremia (35%), pneumonia (28%), and sepsis (12%), similar to frequencies reported with IC without VEN. 

VEN was also assessed in combination with cladribine, high-dose cytarabine, and idarubicin (CLIA + VEN) in patients with ND-AML [33]. Forty-one patients received CLIA + VEN, and nine patients received CLIA + VEN + FLT3I (gilt). The composite CR rate was 94% (*n* = 47/50). MRD-negative remissions using MFC was 82% (*n* = 37/45). Twelve month OS was 85%respectively. Common grade 3 or greater AE’s included febrile neutropenia (74%), diarrhea (4%), other infection (6%) 

A phase 2 trial also explored VEN in combination with standard 7 + 3 induction [34]. Thirty-three ND-AML patients with a median age of 40 (30–47.5) participated in the study. Composite CR was attained in 91%. Of the patients who achieved CRc, 97% had undetectable MRD (measured using multiparameter flow cytometry). Grade 3 or greater AEs included sepsis (12%), pneumonia (21%), and febrile neutropenia (55%).

## 6. Venetoclax Combined with Targeted Therapy: Doublet Combinations

### 6.1. Cobimetinib + VEN

Pre-clinical data demonstrated the combination of VEN and the MEK1 inhibitor cobimetinib showed synergistic antileukemic effects. The combination was initially studied in 11 AML cell lines. A total of 7 out of 11 cell lines exhibited synergistic growth inhibition. In addition, synergy was noted in VEN and cobimetinib-resistant cell lines [26]. 

Given the pre-clinical data supporting the combination of VEN with cobimetinib, the combination was tested in a phase 1b trial, including 22 patients with R/R-AML. In this high-risk patient population (median age 72 years [range 60–93] and median number of prior therapies: 2 (1–10) the combination resulted in a CR/CRi rate of 18% (N = 4/22). The duration of response ranged from 1–5 months [35].

VEN + cobimetinib was difficult to tolerate, resulting in frequent AE’s Grade 3 or greater AEs included diarrhea (36%), febrile neutropenia (23%), fatigue (9%), decreased appetite (5%), and hypokalemia (5%). Three deaths occurred secondary to sepsis, pneumonia, and respiratory failure, respectively. While targeting MCL-1 and BCL2L1 represents an attractive therapeutic combination, cobimetinib + VEN appeared to have modest clinical activity with significant toxicity, limiting the combination’s clinical application in AML.

### 6.2. Idasanutlin + VEN

Idasanutlin, a second-generation, orally available small molecule MDM2 antagonist, combined with VEN demonstrated manageable safety and encouraging activity in older, unfit patients with AML. An open-label phase 1b trial investigated idasanutlin + VEN in 55 patients with R/R-AML (N = 50) or secondary AML (sAML; N = 5) [36]. Fifty patients were enrolled in the dose escalation, and six patients were enrolled in the dose optimization stage. Most patients had received a median of one prior line of therapy; a minority (4%) underwent prior HCT. VEN was administered daily and idasanutlin daily or twice daily on days 1–5 every 28-day cycle. 

CRc (CR + CR with incomplete hematologic recovery (CRi) + CR with incomplete platelet recovery (CRp)) was attained in 26% of patients. Responses occurred early (median time to best response was 1.4 months) with a median duration of response of 3.9 months. After a relatively short median follow up of 4 months, the median OS was 5.1 months (13.8 months in patients attaining a CRc). 

Of interest, MDM2 expression was not associated with a response. A trend toward improved antileukemic activity was observed in patients with a high (defined as >1.5) vs. low (defined as ≤1.5) ratio of baseline BCL2:BCL2L1 or BCL2:MCL1 ratio (52.6% vs. 23.5%, *p*-value: 0.097—percent of patients responding). Patients with mutations in *IDH1/2* and/or *RUNX1* appeared to benefit most from idasanutlin + VEN, with the longest median OS observed in patients with *IDH1/2* mutations (median: 7.64 months). Only 30% (N = 3/10) of patients with *TP53*-mutated AML responded to idasanutlin + VEN, including two patients with co-occurring *IDH1* and *RUNX1* mutations, respectively. Given the dependency for intact TP53 for idasanutlin activity, the low response rate in patients with *TP53*-mutated AML was not unexpected. Similar to AZA + VEN, *TP53* (33.3%) and *RAS* pathway gene mutations (27.8%) were frequently observed mechanisms of resistance.

Common AEs following idasanutlin + VEN included diarrhea (87.3%), nausea (74.5%), vomiting (52.7%), hypokalemia (50.9%), and febrile neutropenia (45.5%). Aside from febrile neutropenia, most AEs were grade 1 or 2. Mandatory antidiarrheal prophylaxis was implemented during dose escalation. Only one patient experienced an AE necessitating treatment discontinuation. Three patients developed tumor lysis syndrome (TLS), including one patient developing grade 3 clinical TLS that resolved with supportive care alone. Mortality at 30 and 60 days was 5.6% and 16.9%, respectively. Three dose-limiting toxicities occurred in the study, including two for neutropenia. The maximum tolerated dose was VEN 600 mg and idasanutlin 200 mg. At the time of analysis, the recommended P2 dose had not been identified. 

Overall, VEN–idasanutlin demonstrated activity in an older, unfit R/R-AML population with few available treatment options. Studies incorporating alternative MDM2 inhibitors (i.e., siremadlin) in the setting of a suboptimal response to AZA + VEN are ongoing (NCT05155709) [37].

### 6.3. Mivebresib + VEN

Miverbresib (MIV) is an orally available small molecule Bromodomain and extra terminal (BET) inhibitor. In animal models of AML, tumor regression has been observed with the use of Miverbresib. In pre-clinical studies, Miverbresib, in combination with VEN, demonstrated more antileukemic activity in AML than either agent alone. A phase 1 study of Mivebresib alone or in combination with VEN in patients with R/R-AML demonstrated an antileukemic effect in both monotherapy and in combination with VEN (NCT02391480). In pre-clinical models, the in vitro activity of Mivrebesib and VEN showed a 100-fold shift in the dose–response curve in a 24 h assay, indicating the combination was more effective at inducing leukemic cell death than either agent in monotherapy [38]. 

Forty-four patients with R/R-AML were enrolled in the study. All patients enrolled in the study had received at least one prior therapy. A total of 19 patients received MIV alone, and 25 patients received MIV in combination with VEN. Five patients in the MIV monotherapy group experienced disease progression and were switched to the MIV + VEN arm after a washout period. 

A total of 36 of the 44 patients were able to be assessed for antileukemic response. Of the 36 patients, 44% (*n* = 16/36) received MIV alone, 42% (*n* = 15/36) received MIV + VEN, and 14% (*n* = 5/36) were switched from MIV alone to MIV + VEN. Among the 30 total patients receiving MIV + VEN, the responses were as follows: 7% (*n* = 2/30) achieved a CR, another 7% achieved partial remission (PR), 7% achieved a morphologic leukemia-free state, 3% had aplasia, and 40% of patients had resistant disease. The median OS in the MIV monotherapy group was 15.7, 22.0, and 12.6 weeks for the 1.5, 2.0, and 2.6 mg doses, respectively. In comparison, the median OS in the MIV + VEN group was 37.4, 11.8, and 11.4 weeks at the 1.0 mg + 400 mg, 1.0 mg + 800 mg, and 2.5 mg + 800 mg dose levels, respectively. 

Common AEs experienced in the MIV monotherapy group included dysgeusia (79%), fatigue (68%), nausea (47%), and diarrhea (47%). Overall, 88% (*n* = 22/25) of patients receiving MIV + VEN and 80% (*n* = 4/5) of switched patients experienced AEs. In the MIV monotherapy group, 74% of patients (*n* = 14/19) experienced serious AEs.

Overall, MIV + VEN demonstrated modest activity in the R/R-AML population in comparison to MIV monotherapy. Studies evaluating the safety and activity of MIV in combination with AZA in R/R-AML patients are currently ongoing [39].

## 7. Venetoclax Triplet Combination Therapies

### 7.1. IVO + VEN +/− AZA

*IDH1* mutations are present in ~7–8% of cases of AML [11]. *IDH*-mutated leukemias are uniquely dependent upon the anti-apoptotic protein BCL2 for survival secondary to 2-hydroxyglutarate mediated lowering of the mitochondrial outer membrane permeability threshold (thereby lowering the induction threshold of the intrinsic apoptotic pathway and subsequent cell death) [40].

The orally available IDH1 inhibitor ivosidenib (IVO) demonstrated efficacy as a single agent in both ND and R/R-AML [11,41], and was evaluated in combination with VEN with or without AZA in a PIb/II study of *IDH1*-mutated myeloid malignancies [42]. 

Thirty-one patients with advanced *IDH1*-mutated myeloid malignancies were treated within four separate cohorts, two of which received VEN 400 mg or 800 mg once daily (D1–14 every 28 days) in combination with IVO 500 mg once daily (administered continuously beginning on C1D15). Two additional cohorts evaluated the triplet regimen of IVO + VEN (again dosed at 400 or 800 mg) combined with AZA 75 mg/m^2^ (D1–7 every 28 days). The median patient age was 67 years (range 44–84). Among patients with AML (comprising 84% of the study population), 63% had ND-AML, and 36% had R/R-AML. 

The ORR for the study population was 94%, with a CRc (CR + CRh + CRi) rate of 87%. In patients with ND-AML that were evaluable for measurable residual disease assessment (MRD) using multiparameter flow cytometry, 86% of patients treated with IVO + VEN + AZA compared with 25% of patients treated with IVO + VEN attained MRD-negative CRc. Combining targeted therapies with IVO + VEN +/− AZA appeared effective at eliminating the *IDH1*-mutated leukemic clone. *IDH1* mutations were undetectable using digital droplet PCR in 64% of patients receiving five or more cycles of treatment.

Median event-free survival and OS were 36 and 42 months, respectively. OS was 71% (ND-AML: 67%, R/R-AML: 50%) at 24 months. In patients with AML, MRD-negative CRc improved OS (median NR vs. 8 months, *p*-value: 0.002). 

Both IVO + VEN and IVO + VEN + AZA were well tolerated. Common AEs (grade 3–5) occurring in ≥10% of patients included febrile neutropenia (29%) and pneumonia (23%), within the range observed with AZA + VEN therapy. Two patients developed TLS (including one episode considered a dose-limiting toxicity); four patients developed IDH-differentiation syndrome (IDH-DS). All episodes of TLS and IDH-DS were manageable and resolved with supportive care alone. Given the safety and efficacy of IVO + VEN + AZA, phase 2 expansion cohorts are enrolling in ND-AML and R/R-AML (NCT03471260). Whether IVO + VEN + AZA improves outcomes compared to the current standard of AZA + VEN will require future prospective, randomized investigations.

### 7.2. ENA + VEN +/− AZA

*IDH2* mutations occur in ~12–15% of patients with AML [43] and are more frequent in older patients [44]. Enasidenib (ENA) is a small molecule mutant IDH2 inhibitor approved as monotherapy in IDH2-mutated R/R-AML [12]. 

ENA combined with VEN in *IDH2*-mutated R/R-AML was evaluated in a small phase I/II study of eleven patients (R/R-AML: N = 10; MDS: N = 1) [45]. Among patients completing a minimum of one cycle of treatment, the ORR was 55% (N = 5/9; CR: 22%, CRi: 33%). ENA + VEN was well tolerated, with no episodes of TLS or IDH-DS observed.

The triplet regimen of ENA + VEN + AZA was subsequently evaluated in a subset of patients from a now-concluded phase 2 study evaluating AZA + ENA (VEN or FLT3i were permitted as clinically indicated) [46]. Twenty-six patients were evaluated (ND-AML = 7, R/R-AML = 19). The median age was 77 in patients with ND-AML and 64 in patients with R/R-AML; 42% of patients with R/R-AML were in the first relapse, while 58% were in the second or later relapse. Seven patients with R/R-AML received ENA + VEN + AZA. 

Though limited by small numbers, ENA + VEN + AZA resulted in a CRc rate of 86% (N = 6/7), including a true CR rate of 71%. Notably, five of the six patients who received ENA + VEN + AZA had prior exposure to HMA therapy (one had prior ENA exposure). After a median follow up of 11.2 months, the median OS was not reached. The reported 6-month OS was 70%.

Treatment-emergent AEs ≤ grade 3 occurred in 85% of patients in both ND and R/R-AML subgroups. The most frequent AEs ≥ grade 3 included febrile neutropenia (23%; ND-AML: N = 1 and R/R-AML: N = 5) and indirect hyperbilirubinemia (35%; ND-AML: N = 2 and R/R-AML: N = 7). Grade 3 or greater febrile neutropenia occurred in 14% (N = 1) of patients with ND-AML and 5% (N = 1) of patients with R/R-AML. Most AEs were manageable without dose interruption. IDH-DS was reported in two patients, successfully managed using corticosteroids with or without temporary interruption of ENA. Thirty- and 60-day mortality rates were 0% and 5%, respectively. 

ENA + VEN + AZA appears safe and active in *IDH2*-mutated AML. Similar to triplet regimens containing IDH1 inhibitors, whether or not this improves outcomes compared to AZA + VEN (where the median OS was not reached in patients with *IDH2* mutations) remains an outstanding question [47].

### 7.3. GILT + VEN +/− AZA 

Activating mutations in *FLT3* occur in approximately 30% of patients with ND-AML and are associated with increased rates of relapse and reduced survival [48,49,50]. Gilteritinib, an orally available type 1 FLT3i active against both *FLT3*-ITD and TKD mutations, demonstrated efficacy compared with salvage chemotherapy in patients with R/R *FLT3*-mutated AML [51]. In pre-clinical models, gilteritinib combined with VEN (VEN + gilt) resulted in synergistic leukemic cell death in both *FLT3*-mutated and FLT3 wild-type AML [52]. 

In *FLT3* wild-type AML, VEN + gilt demonstrated a synergistic antileukemic effect. The combination increased cell death by decreasing the phosphorylation of ERK and GSK3B through combined AXL and FLT3 inhibition and suppressing MCL-1 (an anti-apoptotic protein). Of particular interest, even within AZA + VEN-resistant cell lines, the combination of VEN + gilt showed activity [53].

The activity of VEN + gilt was investigated in a phase Ib, open-label, dose-escalation, and expansion study in R/R-AML. The study population included 61 patients (median age: 63 years [range: 21–85]) who had previously received one or more treatments (median: two) [54]. Most patients had either intermediate (56%) or poor (34%) risk AML defined by cytogenetics. A total of 92% of patients had *FLT3*-ITD (72%), *FLT3*-TKD (15%), or both ITD/TKD mutations (5%); 64% percent (N = 36/56) received prior FLT3i and 10% received prior VEN treatment. VEN + gilt treatment consisted of 400 mg oral VEN and 80 mg or 120 mg oral gilteritinib once daily. 

The modified composite complete response (mCRc) rate (complete response (CR) + CR with incomplete blood count recovery (CRi) + CR with incomplete platelet recovery (CRp) + morphologic leukemia-free state (MLFS)) among patients treated at any dose was 75%, which is notable in this heavily treated population with significant exposure to prior FLT3i. VEN + gilt induced deep molecular responses, with 66% of patients achieving *FLT3*-ITD variant allele frequency levels < 10^−2^. Another 12% attained undetectable *FLT3*-ITD levels. After a median follow up of 17.5 months, the median OS in patients without vs. with prior FLT3i exposure was 10.6 vs. 9.6 months, respectively. Median OS was 6.7 months in patients with prior VEN exposure and 8.8 months in patients who had undergone prior allogeneic hematopoietic cell transplant (HCT).

Myelosuppression was common with VEN + gilt. Grade 3/4 cytopenias occurred in 80% of patients. Grade 3/4 thrombocytopenia and neutropenia occurred in 34% and 31% of patients, respectively. SAEs occurred in 75% of patients, most commonly febrile neutropenia (44%) and pneumonia (13%). AEs prompted dose interruption of VEN or gilt in 51% and 48% of patients, respectively. A total of 15% of study participants discontinued VEN due to AEs, and 13% discontinued gilt. The reported 30 and 60-day rates were 0% and 13%, respectively. Based on the observed efficacy and toxicity profile, the recommended dose was 400 mg of VEN with 120 mg gilt administered once daily in 28-day cycles. 

VEN + gilt led to a response in a significant portion of patients with R/R-AML following prior FLT3i and/or VEN therapy, highlighting the combination activity in this difficult-to-treat population. The activity in R/R-AML warrants investigation of the doublet as a frontline treatment for ND-AML. However, this efficacy must be carefully balanced against the observed myelosuppression and infection risk. Dose modifications for the management of cytopenias, appropriate patient monitoring, transfusion support, and prompt treatment of neutropenic fever are requisite when using VEN + gilt treatment. 

While VEN + gilt is highly active in R/R-AML, it remains unknown if triplet combinations can further improve outcomes in this high-risk molecular subgroup. A prospective phase I/II study of AZA + VEN + gilt in patients with *FLT3*-mutated AML reported impressive results in the frontline setting [55]. In 21 patients with *FLT3*-mutated ND-AML, 95% (N = 20/21) of patients attained a CR, and 5% attained an MLFS corresponding to an ORR of 100%. After a median follow up of 10 months, a median OS was not reached (estimated 12-month OS: 80%).

Given the myelosuppression observed with VEN + gilt, a bone marrow examination was performed early (C1D14), demonstrating all patients either achieved morphologic remission or had insufficient/aplastic marrow highlighting the potency of AZA + VEN + gilt. To mitigate AEs related to myelosuppression, VEN and gilt were held on C1D14 in responding patients to permit hematologic recovery; VEN was subsequently reduced to 7 days for cycle 2 onward in responding patients. With this judicious medical management, no 30 or 60-day mortality was observed. 

As anticipated, responses were more modest in patients with R/R-AML, especially those with prior gilteritinib and/or HMA + VEN exposure. A total of 21% (N = 4/19) of patients achieved a CR, 16% (N = 3/19) achieved CRi, and 37% (N = 7/19) achieved an MLFS. After a median follow up of approximately 24 months, the median OS was 5.8 months in the overall R/R-AML population and 10.5 months in patients not treated with prior HMA + VEN or gilt. An alternative triplet combination using decitabine combined with VEN + gilt is also currently under investigation (NCT03013998). 

### 7.4. AZA + VEN + Magrolimab

CD47/SIRP*α* functions as an anti-phagocytic signal and is overexpressed in multiple malignancies [56,57]. Increasing CD47 expression correlates with inferior outcomes in AML. Conversely, favorable risk AML is associated with lower CD47 expression [58]. 

Magrolimab (an anti-CD47 monoclonal antibody) combined with AZA + VEN was evaluated in a phase I/II study [59]. Patients received standard AZA + VEN in combination with magrolimab administered as 1 mg/kg on C1D1 and C1D4, 15 mg/kg on C1D8, 30 mg/kg on C1D11, C1D15, and C1D22, 30 mg/kg weekly in cycle 2, and 30 mg/kg biweekly from cycle 3 onward. This ramp-up dose was used to mitigate on-target hemolytic anemia observed with magrolimab. Seventy-four patients were enrolled (ND-AML: N = 41 and R/R-AML: N = 29). Within the ND-AML cohort, 32 patients had de novo and 9 had untreated secondary AML. Twenty-seven patients with ND-AML had *TP53* mutations.

A total of 80% of patients (N = 33/41) with ND-AML responded, including 74% of patients with *TP53*-mutated AML (ORR in *TP53* wild-type patients was 93%). Corresponding CR + CRi rates for *TP53*-mutated and *TP53* wild-type patients were 63% and 86%, respectively, which compares favorably to previous findings of AZA + magrolimab in *TP53*-mutated MDS. Only 12% of patients (N = 2/17) with R/R-AML and prior VEN exposure attained a response, and this cohort was closed for futility. Conversely, in patients with R/R-AML who were VEN naïve, 67% (N = 8/12) responded. 

After a median follow up of 9.2 months, median OS was not reached in the ND-AML cohort (estimated 1-year OS for *TP53*-mutated and *TP53* wild-type patients was 53% and 83%, respectively). OS was more modest for patients with untreated secondary AML (median OS: 7.2 months) or R/R-AML (median: 7.4 months). 

AZA + VEN + magrolimab was well tolerated. The most common AEs were febrile neutropenia (50%), pneumonia (38%), hyperbilirubinemia (11%), transaminitis (11%), creatinine elevation (8%), and hypokalemia (8%); 24% of patients had ≥grade 3 anemia. Mortality at 4 and 8 weeks was 0% and 7%, respectively [60]. These results are encouraging (particularly for *TP53*-mutated AML), and results of confirmatory ongoing phase 3 trials are eagerly anticipated (NCT05079230).

### 7.5. AZA + VEN + Eprenetapopt

Eprenetapopt, a small-molecule p53 reactivator, was evaluated in combination with AZA + VEN in a phase 1 study enrolling 49 patients with de novo or secondary AML harboring at least one pathogenic *TP53* mutation [61]. Treatment consisted of 4.5 g/day of eprenetapopt (4.5 g D1–D4) in addition to standard AZA + VEN dosing. A total of 43 patients enrolled in the triplet combination cohort were evaluable for safety and 39 for activity. 

The ORR was 64% (N = 25/39; CR + CRi: 56%; MLFS: 3%; PR 5%). Most patients attained their best response within two treatment cycles. The median duration of response was 4.2 months. After a median follow up of 10.3 months, the median OS was 7.3 months. The Median OS was not reached in a landmark analysis performed at 28 days in patients attaining CR. The median OS in patients receiving HCT in remission was 20 months. 

AZA + VEN + eprenetapopt demonstrated a safety profile similar to AZA + VEN. Common AEs ≥ grade 3 include febrile neutropenia (47%), thrombocytopenia (30%), anemia (22%), leukopenia (24%), and anemia (20%). No dose-limiting toxicities were encountered in the study. While the eprenetapopt in combination with AZA + VEN is tolerable, whether or not the triplet regimen represents an improvement over AZA + VEN (CR/CRi rate of 55.3%; median OS ~ 5–7 months in *TP53*-mutated AML) [4,62] will require randomized investigations. Additional ongoing trials of VEN triplet combinations are displayed in Table 1.

## 8. Conclusions

These preliminary studies utilizing VEN doublets or triplets combined with additional targeted therapeutics and hypomethylating agents represent welcome additions to the treatment of AML. It is worth recognizing many of these aforementioned studies are early-phase, single-arm investigations and are limited by smaller sample sizes. While compelling, future trials encompassing larger patient populations and, ideally, randomized controlled trials, if feasible, are required to validate these early results. 

An area of concern with VEN combinations is the degree of myelosuppression and infectious complications. Myelosuppression was common across combination studies, particularly when combining VEN with FLT3i. Future investigations defining the optimal dose and duration of included agents (including attenuating the VEN duration to 7 or 14 days per cycle) to mitigate myelosuppression and resultant infectious AEs remain paramount as more studies utilize a VEN or HMA + VEN backbone. 

The development of VEN combination therapies is promising. However, VEN resistance remains a challenge. While patients with *FLT3* or *IDH1/2* mutated AML have encouraging outcomes with VEN combinations, treatments targeting other molecular subgroups (i.e., *TP53*, *N/KRAS* mutated) and inhibitor combinations (i.e., MCL1 inhibitors which have demonstrable cardiac toxicity) require further optimization and study. 

Importantly, VEN doublets and triplets appear efficacious and tolerable, facilitating their use in older patients traditionally unable to receive IC. Indeed, the activity is impressive enough that AZA + VEN (NCT04801797) and IC + VEN combinations are now being tested in younger patients as well [32,33]. Given these recent advancements, progress in AML treatment is indisputable. As patients experience longer survival in a disease where historically, few older patients lived beyond 12 months, investigations with VEN-based therapy will need to account not only for initial efficacy, but also the tolerability and durability of complex treatment regimens, which will undoubtedly help to improve the quality and quantity of life for patients living with AML. 

## Figures and Tables

**Figure 1 cancers-15-03589-f001:**
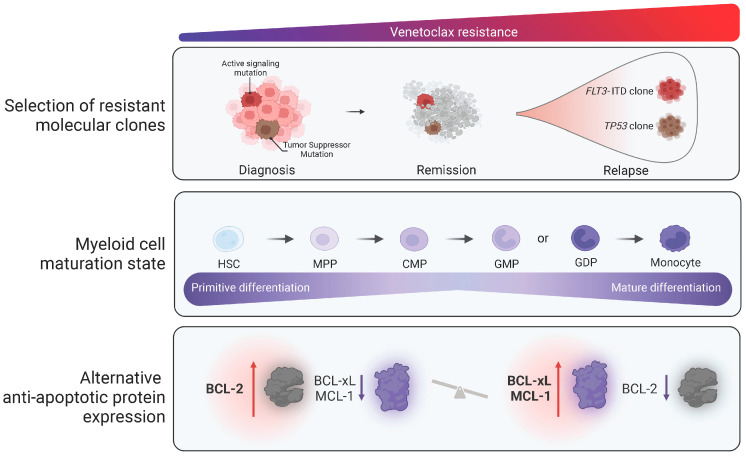
Currently proposed mechanisms of resistance to venetoclax in AML (increasing venetoclax resistance is demonstrated moving from left to right of the figure). Mechanisms associated with resistance to venetoclax include primary or secondary acquisition of molecularly defined resistant subclones, myeloid maturation state, and increased levels of alternative anti-apoptotic proteins, including BCL2L1 and MCL1, with respect to BCL-2 levels.

**Figure 2 cancers-15-03589-f002:**
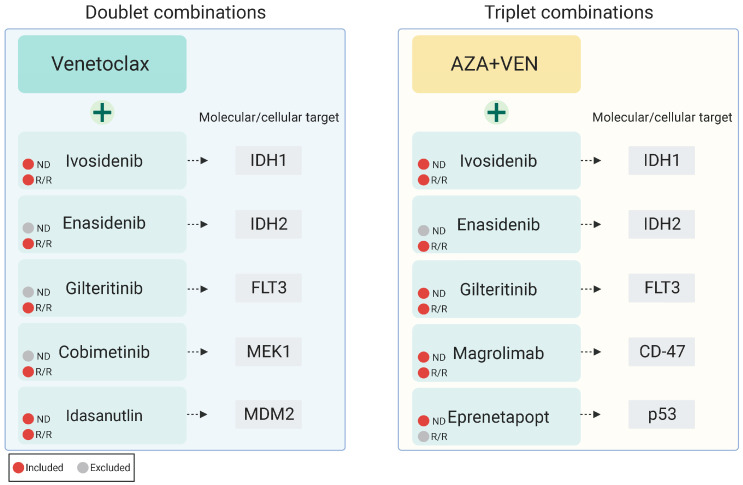
Published investigations of venetoclax-based targeted combination therapies patients with newly diagnosed or relapsed/refractory AML.

**Table 1 cancers-15-03589-t001:** Ongoing clinical trials of venetoclax triplet combination therapies.

Combination Therapy	National Clinical Trial Number
Venetoclax + Azacitidine + Lintuzumab	NCT03932318
Venetoclax + Azacitidine + Siremadlin	NCT05155709
Venetoclax + Azacitidine + SL-401	NCT03113643
Venetoclax + Azacitidine + Tamibarotene	NCT04905407
Venetoclax + SNDX-5613 + ASTX727	NCT05360160
Venetoclax + Azacitidine + Pembrolizumab	NCT04284787
Venetoclax + Decitabine + Quizartinib	NCT03661307
Venetoclax + Uproleselan + Azacitidine	NCT04964505
Venetoclax + ASTX27 + Gilteritinib	NCT05010122
Venetoclax + Azacitidine + Trametinib	NCT04487106
Venetoclax + Azacitidine + Ivosidenib	NCT03471260
Venetoclax + Decitabine + Gilteritinib	NCT03013998
Venetoclax + Azacitidine + Magrolimab	NCT05079230
Venetoclax + Intensive chemotherapy	NCT04801797
Venetoclax + Azacitidine + Evorpacept	NCT04755244
Venetoclax + Azacitidine + Revumenib	NCT05761171
Venetoclax + Azacitidine + Pevonedistat	NCT04266795
Venetoclax + Azacitidine + CC-90011	NCT04748848
Venetoclax + Azacitidine + IMGN632	NCT04086264

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
