# Peer review of "Building on Foundations: Venetoclax-Based Combinations in the Treatment of Acute Myeloid Leukemia"

_cancers, 2023, doi:10.3390/cancers15143589_

Round 1

Reviewer 1 Report

1. In the Introduction section, please specify the responses to intensive chemotherapy in the 60-65 year-old and the 65-74 year-old groups, highlighting the inferior responses that might be related to the underlying adverse disease biology.

2. It will be beneficial to use a diagram to illustrate the mechanisms of action of the various small molecular inhibitors.

3. There is no mention of combinations of venetoclax with intensive chemotherapy in the frontline and relapsed/refractory setting. Examples include VEN + “5+2” in newly diagnosed AML, and VEN + FLAG-Ida in relapsed/refractory AML.

4. Please discuss inn each section the mechanisms of synergism between venetoclax and other small molecules.

5. Please include a section discussion the combination of venetoclax with azacitidine highlighting the data from VIALE-A and VIALE-C studies. In the subgroup of patient with NPM1-mutated AML achieving a molecular response, prolonged remission was observed. Please include and discuss this point.

acceptable with minor amendments

Reviewer 2 Report

The authors provide a well written and up-to-date review of venetoclax-based lower-intensity targeted therapies in patients with treatment-naive and relapsed/refractory acute myeloid leukemia.

Comments/Queries:

1. In the section Venetoclax resistance (page 3, lines 100-128), the authors need to distinguish resistance (primary or secondary) to single agent VEN versus resistance (primary or secondary) to VEN-based therapies, e.g. resistance to VEN + AZA (reference 33 and 38) which is included in the section. The authors also provide more discussion on the rationale for the VEN-based combinations, especially with respect to how they overcome VEN resistance  

2. In the section Venetoclax Doublet Combination therapies, the subsection Gilteritinib + VEN (pages 5 and 6, lines 152 - 193) should be combined with the subsection AZA + VEN + gilteritinib (pages 8-9, lines 301 - 322) as this would be more consistent with the other subsections in the category (i.e. IVO + VEN +/- AZA and ENA + VEN +/- AZA)

3. In section Venetoclax Doublet Combination therapies (pages 5-7), the phase 1 study of VEN + the BET inhibitor mivebresib should be included (even if not impressive results, as this informs the readers of other combinations and may provide information as to the reason for lack of response). Borthakur et al. Cancer 2021; 127(16):2943-53

4. Table 1 (pages 10 and 11), should include the following triplets: (a) VEN + AZA + evorpacept; (b) VEN + AZA + revumenib; (c) VEN + AZA + pevonedistat; (d) VEN + AZA + CC-90011 and (e) VEN + AZA + IMGN632

5. On page 1, line 45 -  the "M" in "Magrolimab" should be a small case "m"

6. On page 2, lines 65-67 - the authors state "FLT3 inhibitors (FLT3i) including midostaurin and gilteritinib improve survival when combined with induction chemotherapy or with VEN (14 - 17)." However, references 15 and 17 refer to the ADMIRAL study with single agent gilteritinib compared with intensive chemotherapy. None of these references include VEN + gilteritinib or VEN + midostaurin. Please provide the appropriate references for this statement and/or modify the statement. 

7. On page 4, the figures for Figures 1 and 2 are switched

8. The reference section needs to be carefully reviewed and changed (there may be others not listed below):

(a) The following references need to be reformated: 28, 40, 44, 47, 50, and 58

(b) There is a duplication of the same references - references 28 and 30 are the same (but formatted differently); similarly references 33 and 38 are the same (but formatted differently)

(c) In reference 33, line 547 - the phrase "Monocytic AML Is Resistant to Venetoclax" needs to be deleted (it is not part of the article's title)

Reviewer 3 Report

My biggest concern is that manuscript focuses on a handful of venetoclax based combinations and I am not sure why only those combinations were discussed. Other than ivosideib, enasidenib, magrolimab, idasanutlin, gilteritinib, epretenapot and cobimetinib, there are multiple other venetoclax based combinations with intensive chemotherapeutic regimens or targeted therapy in ongoing clinical trials. Some of those studies have already reported prelim results. Also, table 1 mentions only triplet and quadruplet combinations, there are doublets as well in clinical trials and not sure why they were not included.

Some other comments/recommendations:

Introduction

Add couple of statements about Decitabine + venetoclax and LDAC + venetoclax which are other therapies that are used in addition to azacitidine and venetoclax.

Line 65-67 the statement states that FLT3 inhibitors combined with induction chemotherapy or VEN improve survival. This statement is not completely true and can be misleading. Midostaurin with intensive chemotherapy improves survival. Midostaurin has not been studied with VEN in a study that improved survival. Gilteritinib with induction chemotherapy has not shown clear benefit with survival. Gilteritinib with VEN is still being studied, which is discussed later in the manuscript- so can be deleted here.

Venetoclax resistance

Acquired resistance has been discussed. Few factors have increased chance of primary resistance and not just acquired resistance that should also be mentioned- e.g. TP53 mutation, MCL-1 expression, monocytic differentiation therapy-related AML.

Figure 1 and Figure 2 need to be switched based on the chronology in the main text.

Figure 1- number of patients mentioned- is that the total number of patients in study? Or is it the number of patients enrolled till the manuscript was written. Either way I do not think the number adds anything there and can be removed.

Figure 2- it is hard to follow the alternative anti-apoptotic protein expression part. It may need some explanation in text or be edited in figure itself to make it clearer.

No major concerns.

Reviewer 4 Report

In their review, Oyogoa et al addressed the issue of venetoclax-based combinations in AML.

The background is overall well-written, the description of available trials is comprehensive and balanced. The figures are very illustrative and educational.

Some minor issued should be addressed.

Minor issues

- Abstract, lines 38-39: the authors state “standard of care treatment is typically venetoclax (VEN) combined with 38 the hypomethylating agent (HMA) azacitidine (AZA)”; I would suggest widening to “hypomethylating agents” as the use of decitabine led to similar results although not confirmed in a prospective controlled setting.

- Line 115: the authors state “Intact p53 protein function is also essential for the activity of VEN”; I would suggest rephrasing this sentence as in the present form, the absence of TP53 mutations seems a requirement event for the obtainment of a response, that is not true. Rather, the observed responses in this setting are virtually always short-term, that is a different concept. 

- The names of genes should be reported in italics.

Round 2

Reviewer 2 Report

I agree with and approve the replies/changes made by the authors.

Author Response

Thank you for your comment

Reviewer 3 Report

1. Line 58, the start of the paragraph sounds abrupt and it is hard to figure out the study you are referring to. I would suggest rephrasing the first statement. For example,

‘In one study by Kantarjian et al., …

‘In one study evaluating the use of intensive chemotherapy in older patients….’

You can edit it your own way. These are just examples.

You could add the reference after the first statement for readers to find out what study you are referring to.

2. Line 67- omit AZA

3. Table 1- use either ‘+’ or ‘,’  for all combinations.
